# Robotization of Three-Point Bending Mechanical Tests Using PLA/TPU Blends as an Example in the 0–100% Range

**DOI:** 10.3390/ma16216927

**Published:** 2023-10-28

**Authors:** Julia Głowacka, Łukasz Derpeński, Miłosz Frydrych, Bogna Sztorch, Błażej Bartoszewicz, Robert E. Przekop

**Affiliations:** 1Centre for Advanced Technologies, Adam Mickiewicz University in Poznań, 10 Uniwersytetu Poznańskiego, 61-614 Poznań, Poland; julia.glowacka@amu.edu.pl (J.G.); frydrych@amu.edu.pl (M.F.); bogna.sztorch@amu.edu.pl (B.S.); 2Faculty of Chemistry, Adam Mickiewicz University in Poznań, 8 Uniwersytetu Poznańskiego, 61-614 Poznań, Poland; 3Faculty of Mechanical Engineering, Bialystok University of Technology, 45C Wiejska, 15-351 Bialystok, Poland; l.derpenski@pb.edu.pl (Ł.D.); b.bartoszewicz@pb.edu.pl (B.B.)

**Keywords:** polylactide (PLA), thermoplastic polyurethane (TPU), polymer blends, automation, robotics, flexural properties, thermal analysis, optical microscopy (MO), AI feeding data

## Abstract

This article presents the development of an automated three-point bending testing system using a robot to increase the efficiency and precision of measurements for PLA/TPU polymer blends as implementation high-throughput measurement methods. The system operates continuously and characterizes the flexural properties of PLA/TPU blends with varying TPU concentrations. This study aimed to determine the effect of TPU concentration on the strength and flexural stiffness, surface properties (WCA), thermal properties (TGA, DSC), and microscopic characterization of the studied blends.

## 1. Introduction

As science and the economy continue to advance, researchers in both academic and industry settings are turning to increasingly sophisticated tools to keep pace with the demands of the field. In this environment, effective management of human resources and the ability to adapt quickly are critical. Research process automation is an approach that utilizes technology to streamline and improve the efficiency of research and experimentation. There are many examples of the use of machines, computer technology, and software to support workflow management, data collecting, simulations, or decision making [1,2,3,4]. This approach can be applied across a variety of scientific fields, from engineering [5] and biology [6] to social sciences [7]. A considerable amount of emphasis is being placed on the creation of novel algorithms that can be conveniently stored in the cloud, which is easy to implement in open networking systems [8]. Networking plays a crucial role in Industry 4.0, serving as a means of connecting physical machines in cyber-physical production systems (CPS) [9]. By doing so, it enhances the flexibility and intelligence of production processes, while facilitating machine-to-machine (M2M) communication [10]. This innovative approach to networking has proven effective in optimizing production processes and fostering the growth of the manufacturing industry through the incorporation of smart technology. However, there are lots of challenges, and ensuring open-source systems’ safety from potential hacker attacks is a significant challenge that needs to be addressed [11,12]. While often associated with industrial production, the automation and robotization of research processes are gaining attention as a way to drive scientific progress and improve research outcomes. The automation of experiments is motivated by several benefits. By automating routine tasks, researchers can focus on more complex work that requires critical thinking. Automation can also reduce human error which depends on several factors, such as skills, habits, operator attention, and understanding of procedures, and increases the accuracy and speed of research processes, resulting in faster access to results by minimizing manual work. Kraber and Endsley analyzed how the level of process automation affects operator behavior and interaction and described the effects of the level of automation (LOA) and adaptive automation (AA) on human performance, situation answers, and workload in a dynamic control task. They exhibited intricate interdependencies between the operator and the automated systems [13]. In this paper, a novel approach is presented for the material fabrication and preparation for mechanical testing. The automation of mechanical tests can be useful for anisotropic materials and objects produced by technologies such as 3D printing, where many properties depend on the unique structure created during manufacturing [14,15,16].

Table 1 and Figure 1 provide a comparison between traditional approaches and the proposed method, which involves automated robotic tools for continuous material characterization. Automation eliminates redundant operations in the complex testing process and reduces the need for staff involvement, resulting in more efficient and accurate results.

Through the utilization of high-throughput measurements, it can ascertain changes in measured parameters with greater precision than traditional measurement techniques and is able to prepare a materials database useful at higher levels of research process automation, which can be useful for machine learning in material science. High-throughput measurement methods are a collection of techniques frequently utilized across a wide range of scientific and industrial disciplines to rapidly gather data or analyze a significant volume of samples or data points in order to enhance the speed and effectiveness of data acquisition or analysis, making them exceptionally valuable in situations where a sizable amount of data must be processed promptly. Usually, research methods in polymer science rely on a few selected measurement points to estimate a general trend in global changes in material properties, which is vulnerable to measurement uncertainty. Estimating a trend curve based on a limited number of data points can be achieved through various means, and referring to a global change in a parameter within a specific range may not always precisely reflect the actual changes in properties (refer to Figure 2). The design of plastic products requires a thoughtful selection of materials for their function. This choice is not straightforward, requiring many factors to be taken into account, so, to meet the expectations of designers, researchers are proposing and testing various decision-making models to facilitate the selection of suitable materials. In their discussion, Muhammed Ordu and Oguzhan Der proposed a simple and efficient hybrid multi-criteria decision-making (MCDM) model of material selection for flexible pulsating heat pipe manufacturing [17]. Mastura M. T. et al. studied the material selection of thermoplastic matrices for naturally reinforced green composites dedicated to the automotive industry using Quality Function Deployment for the Environment (QFDE) [18]. The approaches presented in both studies exhibit a better understanding of new perspectives in sustainable material selection by providing mathematical methods for material evaluation. High-throughput experiments can be widely used to explore materials’ parameters across the endless number of composition combinations to support mathematical calculation.

The approach to materials research presented here is well suited for the study of polymeric materials. In recent years, there has been a focus on developing new materials from renewable sources to address the raw material crisis in engineering materials and move toward sustainable solutions for the circular economy [19,20,21]. An increasing number of products in the market are being developed using materials obtained from renewable sources [22]. These materials can be categorized into four distinct groups based on their source and the techniques used to acquire them. These groups include polymers derived from agro-cultural sources (starch-, soy-, and cellulose-based materials), those of animal origin (chitin-based materials), those produced by bacteria (PHA), and polymers obtained through biotechnology (PCL, PLA, PHB) [23,24]. Of particular interest is polylactide (PLA), a biodegradable thermoplastic polyester derived from renewable starch raw materials obtained conventionally by biotechnology methods which are constantly being improved to convert production processes to low- or zero-carbon and to be independent of fossil energy sources [25]. PLA has many advantages and is similar to conventional petrochemical materials in terms of transparency and ease of processing. However, it is often too brittle and stiff, with low strain at break, which limits its use in many applications [26,27,28]. Despite this, PLA is a potential alternative to current petroleum-based polymers, the production of which is closely linked to depleting oil reserves and should be restricted in the future. One solution is to blend PLA with other polymers that have higher flexibility. This is a simple and economically beneficial way to shape the properties of polymeric materials.

Thermoplastic polyurethanes are a type of thermoplastic multi-block copolymers that combine the properties of rubber and traditional thermoplastics. TPUs were introduced by Lubrizol Engineered Polymers to the global polymer materials market in the late 1950s and early 1960s, representing a stage in the evolution of polyurethane material technology [29]. TPUs have unique properties that are a direct result of their chemical structure. TPUs consist of both rigid diisocyanine segments (hard segments, HS) that can organize themselves and act as cross-linking agents, as well as amorphous domains that are derivatives of polyesters and polyester-based polyols, responsible for their elastic behavior (soft segments, SS) [30]. With the ability to control the ratio, structure, or molecular weight of the reaction compounds during the production of thermoplastic polyurethanes, an extensive array of unique TPUs can be synthesized. This allows for precise tailoring of the polymer structure to obtain the desired final properties of the composition [31,32,33]. They can be also produced from bio-based components, which make them more suitable for the future perspective [34,35]. TPUs are widely used due to their physicochemical characteristics, such as flexibility, abrasion resistance, tensile strength, chemical resistance, and good adhesion to various surfaces (coatings, electronics, sensors, textile, adhesives, 3D printing) [36,37,38,39,40]. Due to its unconventional properties including highly elasticity, TPUs can be used to modify other polymeric materials [41]. There has been significant attention in the literature on using thermoplastic polyurethanes as flexible phases in blends with other polymeric materials such as PS [42], PP [43], PMMA [44], and PBT [45]. PLA/TPU blends rheological, mechanical, and thermal properties, as well as shape–memory behavior, and its compatibilization methods were briefly studied by other researchers [46,47,48,49,50].

The use of high-throughput methods has enabled accurate tracing of the process of changing specific material characteristics, representing a new and different approach to the characterization of polymer properties. It sheds new light on issues related to the evaluation of polymer blends. The present study reports the continuous preparation of PLA/TPU polymer blends, varying TPU concentration from 0–100%, and their characterization using high-throughput methods. The study utilized an automated robotic three-point bending measurement rig, specifically developed for these tests. The approach enabled a comprehensive tracing of the formation of blend properties during bending loading (three-point bending). It also determined the nature of the change in hydrophobic–hydrophilic properties of the blends obtained, carried out the microscopic evaluation of the changes in blend structure (MO), and performed TGA analysis.

## 2. Materials and Methods

### 2.1. Materials 

Polylactide (PLA) Ingeo 2003D type was purchased from NatureWorks (Minnetonka, Minneapolis, MN, USA). Thermoplastic polyurethane (TPU) Ravathane 130 D55 type was produced by Ravago Petrokimya Üretim A.Ş. (İzmir, Türkiye).

### 2.2. Preparation of PLA/TPU Blends 

The PLA/TPU blends were prepared by mixing PLA and TPU pelts directly on an Engel e-victory170/80 injection molding machine. Table 2 shows the injection molding parameters.

The mold temperature was maintained at room temperature. A holding pressure with a linear increment over time was applied. Standardized specimens for flexural testing in accordance with PN-EN ISO 20753:2019-01 [51] were fabricated. Beams for automated robotic flexural tests were injected in a continuous feed system with a dosage accuracy of 1% in order to obtain a comprehensive characterization of the material over a range of TPU concentrations from 0 to 100%.

### 2.3. Characterization Methods 

For automatized robotic flexural tests, standardized type B specimens were used. Tests were performed using an MTS Insight testing machine. The servo-mechanical testing machine allows experimental testing under axial loads of up to 1 kN and an elongation/flexural range of up to 750 mm. The traverse speed for measurements was set at 2 mm/min. The measurement was carried out until a deflection arrow of 15 mm was achieved or the specimen broke. The automatic test was conducted continuously over a concentration range of 0 to 100% TPU content in the polymer blends. A total of 628 samples were fluently tested. 

Thermogravimetry (TG) was performed using a NETZSCH 209 F1 Libra gravimetric analyzer (Selb, Germany). Samples of 5 ± 0.2 mg were cut from each granulate and placed in Al_2_O_3_ crucibles. Measurements were conducted under nitrogen (flow of 20 mL/min) in the range of 20–1000 °C and at a 10 °C/min heating rate.

Differential scanning calorimetry (DSC) was performed using a NETZSCH204 F1 Phoenix calorimeter. Samples of 6 ± 0.2 mg were placed in an aluminum crucible with a punctured lid. The measurements were performed under nitrogen in the temperature range of −50–200 °C and at a 10 °C/min heating rate.

Surface topography was analyzed under Digital Light Microscope Keyence VHX 7000 with a 100× to 1000× VH-Z100T lens (Osaka, Japan). All of the pictures were recorded with a VHX 7020 camera. 

Contact angle analyses were performed by the sessile drop technique at room temperature and atmospheric pressure, with a Krüss DSA100 goniometer. Three independent measurements were performed for each sample, each with a 5 µL water drop, and the obtained results were averaged to reduce the impact of surface nonuniformity.

The statistical methods used for three-point bending and water contact angle measurements involved fitting trend curves and determining the coefficient of determination R^2^, using OriginPro v2016 software’s analytical tools. The software was used to enter data obtained from the measurements. A polynomial regression model was selected for three-point bending, and a linear regression model was chosen for WCA measurements. The software automatically fitted the regression model to the data entered, finding the values of the slope coefficient of the curve (slope) and the *Y*-axis intersection point (intercept) that best matched the data entered. After fitting the model, the program calculated the values predicted by the model for all points, and the differences between the actual data values and the values predicted by the model. Subsequently, the sum of the squares of these differences was calculated, which is called the Sum of Squares of Errors (SSE), as well as the Sum of Total Squares (SST), which is the sum of the squares of the differences between the actual data values and their mean. From the SSE and SST values, Origin Pro calculated the coefficient of determination R^2^ using the formula:R^2^ = 1 − (SSE/SST),(1)

### 2.4. Experimental Workstation Setup Design for Automated Robotic Flexural Tests

A special test ring was set up to automate flexural tests. The test stand is built with four main components (Figure 3): Dobot Magician manipulator (1), linear slide (2), sample magazine (3), and MTS Insight testing machine (4). The Dobot Magician manipulator’s task is to take a sample from the magazine equipped with an automatic feeder and transfer it to the testing machine. After taking the sample from the magazine, the manipulator positions itself near the grip of the testing machine and waits for a ready signal. When it receives the signal, it feeds the sample into the machine chuck and withdraws the gripper. It then sends a signal to initiate the start of the testing by the testing machine. During the test, the robot takes another sample. The automatic feeder that is part of the sample magazine constantly monitors the cell from which the sample is taken so that it is filled all the time. In addition, during the stage of feeding the sample to the testing machine holder, the sample already used is pushed by the robot through an appropriately shaped gripper. Such a solution greatly improves the process of automating the exchange of samples during testing. The above-described process is performed automatically until the sample magazine is completely emptied (see in Appendix A).

## 3. Results and Discussion

### 3.1. Flexural Behavior Analysis

Figure 4 presents an analysis of the behavior of PLA/TPU blends under three-point bending. This study involved collecting 628 data points to determine the trend of bending properties as they relate to changes in TPU mass content. Polynomial trend curves were created to evaluate the strength parameters, and both mechanical properties assessed had a coefficient of determination R^2^ ≈ 1, indicating a correct fit of the regression model to the actual data. The flexural strength (Figure 4) and modulus of stiffness (Figure 5) are shown in separate graphs. Both parameters decrease, which can be attributed to the change in the material’s nature from a typical thermoplast to an elastomer.

### 3.2. Thermal Analysis Results

Thermogravimetric analysis is a widely recognized technique that offers insight into the thermal degradation of polymeric materials across a wide temperature spectrum. As part of this study, TGA was employed to monitor the pyrolysis process (decomposition within a nitrogen environment) of PLA/TPU blends for a range of representative samples. This approach enabled the researchers to ascertain the materials’ thermal stability, with the findings presented in both Figure 6 and Table 3. The thermal decomposition of PLA proceeds in one stage, while the decomposition of TPU is more complex and consists of two stages; the first stage involves the decomposition of hard segments of TPU, while the second stage involves the decomposition of soft segments.

The results obtained are in agreement with the literature data [52], which indicate the good quality of the blends obtained and the preservation of compositional constancy in the continuous dosing process carried out. However, the blends are less thermally stable than PLA, as evidenced by the shift in the DTG curves’ extremes to lower temperatures. Exposure to high temperatures can cause transesterification reactions, leading to the weakening of TPU areas in the blends and acceleration of thermal degradation. For blends with 25% TPU, there is a peak dilution phenomenon due to overlapping thermal effects associated with the degradation of both PLA and TPU. This phenomenon disrupts the degradation peak of PLA, leading to its lower intensity due to the division of the energy required to degrade both polymers.

The DTG curves indicate various temperatures related to the decomposition process, including the temperature at which 5% of the mass is lost (T_5%_), the temperature at which the decomposition starts (T_1onset_, T_2onset_), and the temperature at which the maximum mass loss rate occurs for each stage of decomposition (T_1max_, T_2max_). As the proportion of TPU in the sample increased, the temperature of 5% mass loss (T_5%_) decreased by around 10 °C. The addition of TPU into PLA makes the blends more vulnerable to thermal decomposition, as evidenced by the significant reduction in T_1onset_ observed in all biphasic systems. In samples containing 40% and 70% TPU, two decomposition stages were recorded with a similar course to that of pure TPU but with a shift to lower temperatures, indicating accelerated decomposition of soft segments of TPU in the presence of PLA (T_2onset_, T_2__max_).

### 3.3. PLA/TPU Blends Microstructure Evaluation—Optical Microscopy Observations (MO)

The images displayed in Figure 7 were captured for a representative range of test specimens after the bending test.

The pictures illustrate the change in surface morphology (Figure 7(A1–E1)) with respect to the TPU content in the specimen composition and the effect of bending stresses, showing the change in the nature of the material failure pattern during bending in the stress concentration area (Figure 7(A2–E2)). The test material’s characteristics change from the typically brittle failure, due to the effects of bending stresses observed in PLA (complete failure of the specimen), to an elastic response, due to stress relaxation after force subtraction, and return to the original form for test objects made of TPU. The PLA breakthrough exhibits characteristic sharp edges and numerous delaminations and microcracks, while the TPU sample shows no signs of stress-induced damage. The dotted line indicates the depth of the area of microstructural changes at the point of force application for the PLA/TPU blends. As previously discussed, the addition of TPU to PLA positively impacts the material’s elasticity, resulting in a smaller area of permanent microstructural changes in the imaged specimens after the bending test. When the material is bent, cracks occur in the upper layers of the specimen, and rounded-edge microcracks may appear due to a loss of fluidity between the PLA and TPU microareas. The damage caused by bending the PLA/TPU blends is indicated by the red arrows. The rough texture of specimens made from the PLA/TPU blends is due to microheterogeneities caused by the limited miscibility of the polymer phases. It appears that the presence of TPU has a consistent effect on the surface appearance of all blends, regardless of its concentration in the polymer system. The surface of the PLA and TPU samples is significantly smoother than that of the blends, allowing the lines of the direction of filling of the molding cavity by the plasticized material to be seen. Macroscopic images of the chosen samples are presented in Figure 8.

### 3.4. Contact Angle Analysis Results

The results of the surface wetting measurements by water, determined using the sessile drop method, for selected measurement samples made from PLA/TPU blends and natural PLA and TPU polymers are presented in Figure 9. The purpose of the measurement was to determine the effect of TPU content in the blends on the hydrophobic–hydrophilic character of the surfaces of the measurement samples. The trend curve and the coefficient of determination R^2^ were also determined. The WCA changes significantly with an increase in the mass content of TPU in the blend, signifying a change in the surface character of the developed materials. The TPU grade used to modify PLA increases the hydrophobic character of the blends with respect to PLA due to the inherent properties of TPU [53]. TPU is composed of polyurethane segments, which are often hydrophobic or have limited ability to form hydrogen bonds with water molecules [54], while PLA has carbonyl hydrophilic groups (C=O) and also hydroxyl groups (-OH) in its structure, which can form hydrogen bonds with water. These promote interactions between PLA and water molecules, making it a more hydrophilic material than TPU.

### 3.5. Differential Scanning Calorimetry (DSC)

Differential scanning calorimetry analysis was conducted on reference samples of PLA and TPU, as well as blends with varying amounts of TPU (10%, 25%, 40%, 55%, 70%, and 90%). Based on the DSC curves, the glass transition temperature (T_g_), cold crystallization temperature (T_cc_), and melting temperature (T_m_) were calculated for both the first and second measurement cycles (Table 4). Figure 10 illustrates the thermograms of the samples during the second heating cycle. The results indicate that the majority of changes occur during the crystallization stage. For pure PLA, the glass transition, crystallization, and melting temperatures are T_g_ = 61.9 °C, T_cc_ = 127.6 °C, and T_m_ = 153.8 °C, respectively. The DSC curves for pure TPU do not exhibit any transitions in the second heating cycle. The DSC measurements conducted during the second cycle indicate that the presence of dispersed TPU phases has an impact on the PLA crystallization process. The T_cc_ value decreased from almost 128 °C (neat PLA) to around 110 °C (PLA/TPU blend). The thermograms illustrate that the cold crystallization peak is more pronounced for the blends, especially for those containing 10% and 25% TPU. According to the literature, this effect can be attributed to TPU, which acts as a crystallization nucleation agent by providing nucleation spots [55]. In blends that contain 90% TPU, the T_g_, T_cc_, and T_m_ signals are barely discernible.

## 4. Conclusions

Our study delves into the intricacies of polymeric materials utilizing high-throughput methods. We have utilized PLA/TPU blends as a case study to showcase the potential of automated robotic measurements in gauging bending properties. Our research has devised a simple measuring station construction that drastically enhances the efficiency of the three-point bending measurement method, introducing a novel way of analyzing materials. Furthermore, we have verified the feasibility of continuous fabrication and characterization of polymeric multi-component materials. Through thermal analysis, we have established that the compositional consistency of PLA/TPU blends produced through continuous dosage methods was upheld. Additionally, we have gathered evidence of the compositional stability of the materials, obtained by tracing the change in the angle of surface wetting by water. Our microscopic analysis of the failure areas of the specimens during mechanical testing allowed us to determine the effect of TPU content on microstructural transformations in standard components made from PLA/TPU blends. The presence of TPU in the composition of the blends significantly increases the ability of the systems to relax stresses, reducing the negative aspects of their influence on the formation of permanent material damage. Our methods provide a solid foundation for the development of a new approach to high-throughput polymer-based material methods.

## Figures and Tables

**Figure 1 materials-16-06927-f001:**
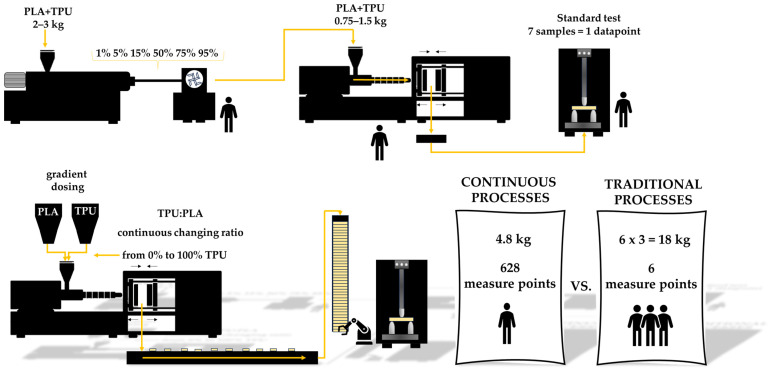
Comparison of traditional and continuous plastics processes.

**Figure 2 materials-16-06927-f002:**
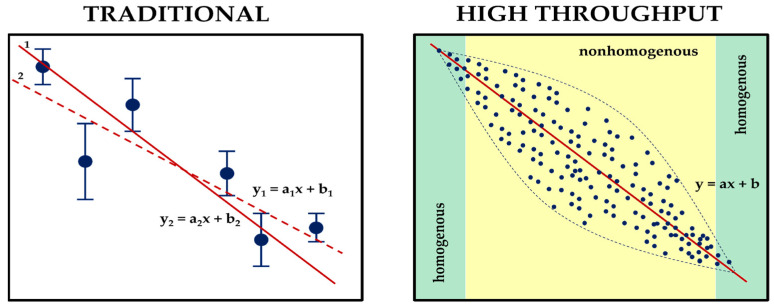
Material characterization ways. In the figure, the blue dots represent the measurement points, while the red lines correspond to the potential trend curves. Each curve’s general equations are displayed, and the figure indicates the indices 1 and 2 for the different curve variants fitted to the traditional measurement data.

**Figure 3 materials-16-06927-f003:**
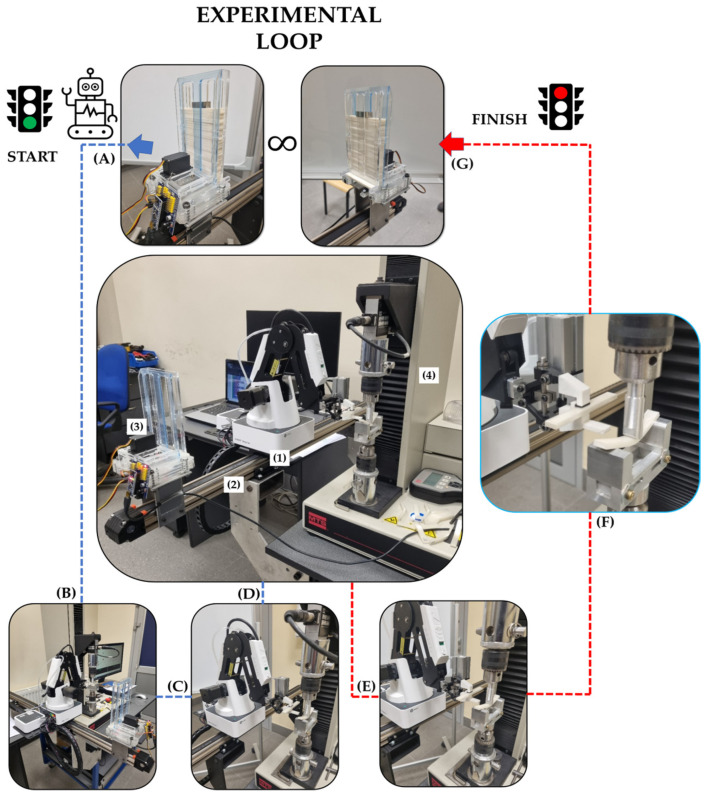
Experimental setup. Experimental steps: (**A**)—taking a sample from the stock (1—roboot, 2—linear slide, 3—sample magazine, 4—testing machine), (**B**–**D**)—robot positioning, (**E**,**F**)—fixing the sample in the measuring holder/removal of the tested sample, (**G**)—end step, taking a new sample.

**Figure 4 materials-16-06927-f004:**
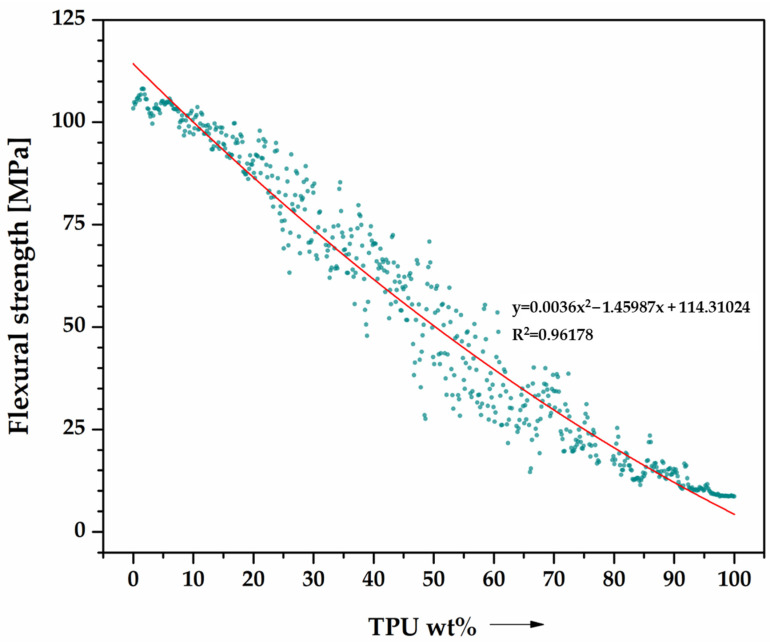
Flexural characteristics of PLA/TPU blends as a function of TPU content (0–100%), performed by the robot (data cloud)—628 data points. Flexural strength.

**Figure 5 materials-16-06927-f005:**
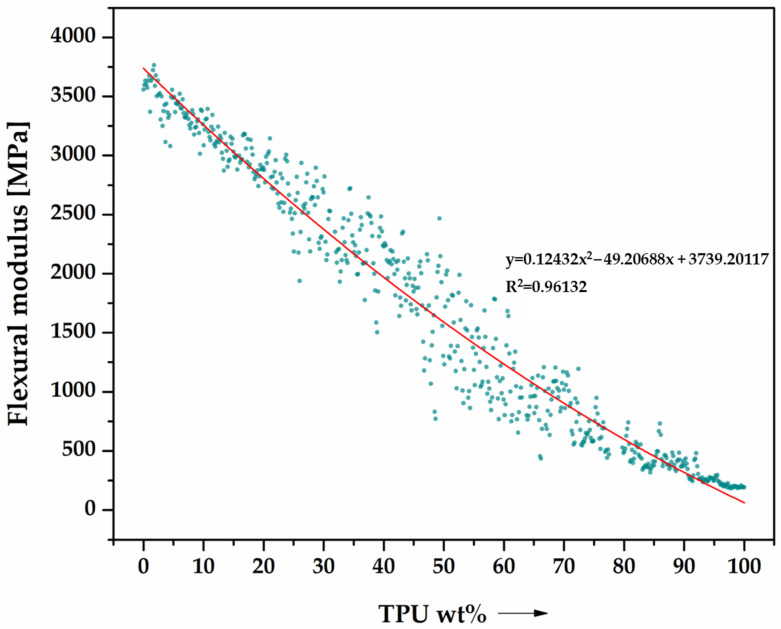
Flexural characteristics of PLA/TPU blends as a function of TPU content (0–100%), performed by the robot (data cloud)—628 data points. Flexural stiffness.

**Figure 6 materials-16-06927-f006:**
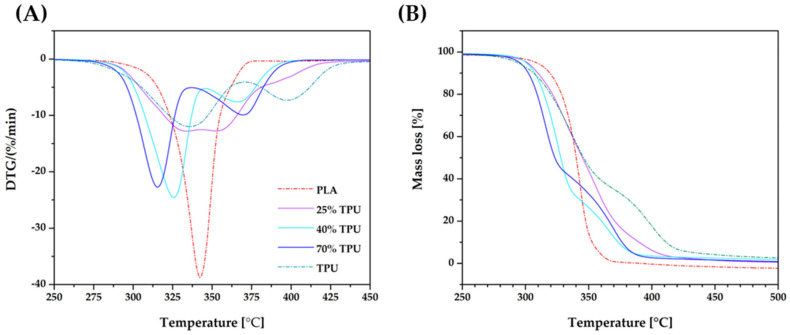
The effects of TPU content on the thermal decomposition of PLA and PLA/TPU blends: DTG (**A**) and TGA (**B**)—N_2_ atmosphere.

**Figure 7 materials-16-06927-f007:**
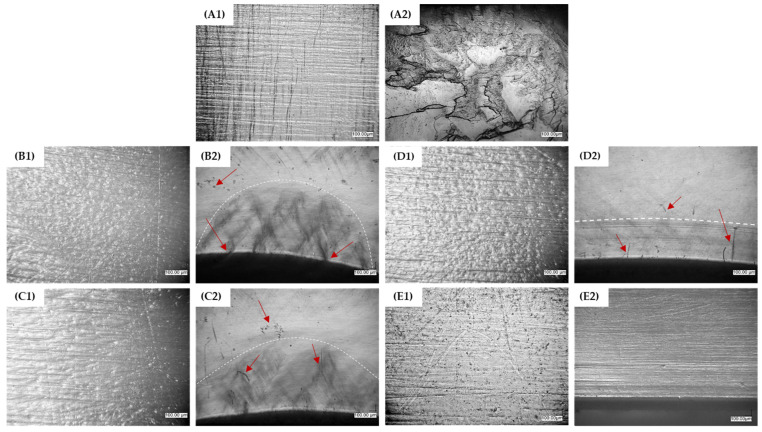
Changes in the surface appearance (**1**) and structure of the samples (**2**) as a result of interaction shearing forces under flexure, from the brittle failure of PLA to the elastic deformation of TPU; (**A**)—PLA, (**B**)—25% TPU, (**C**)—40% TPU, (**D**)—70% TPU, (**E**)—TPU (magnification 100×).

**Figure 8 materials-16-06927-f008:**
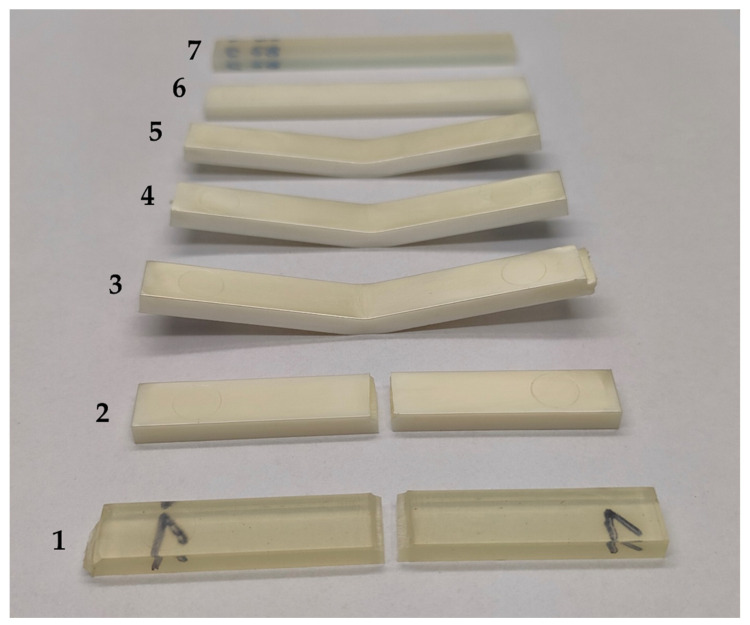
Macroscopic images of selected samples: 1—PLA neat, 2—10% TPU, 3—25% TPU, 4—40% TPU, 5—70% TPU, 6—90% TPU, 7—TPU neat.

**Figure 9 materials-16-06927-f009:**
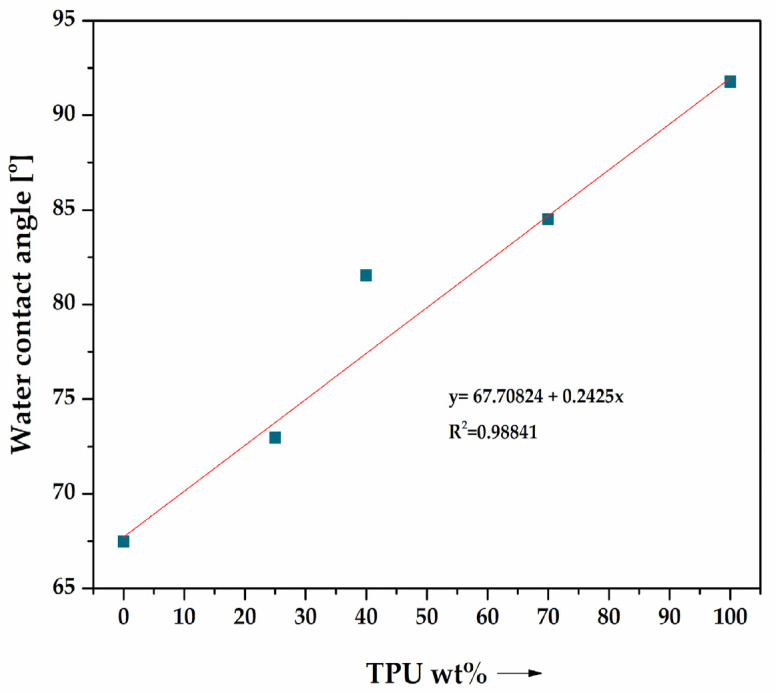
Water contact angle [°] of PLA/TPU blends.

**Figure 10 materials-16-06927-f010:**
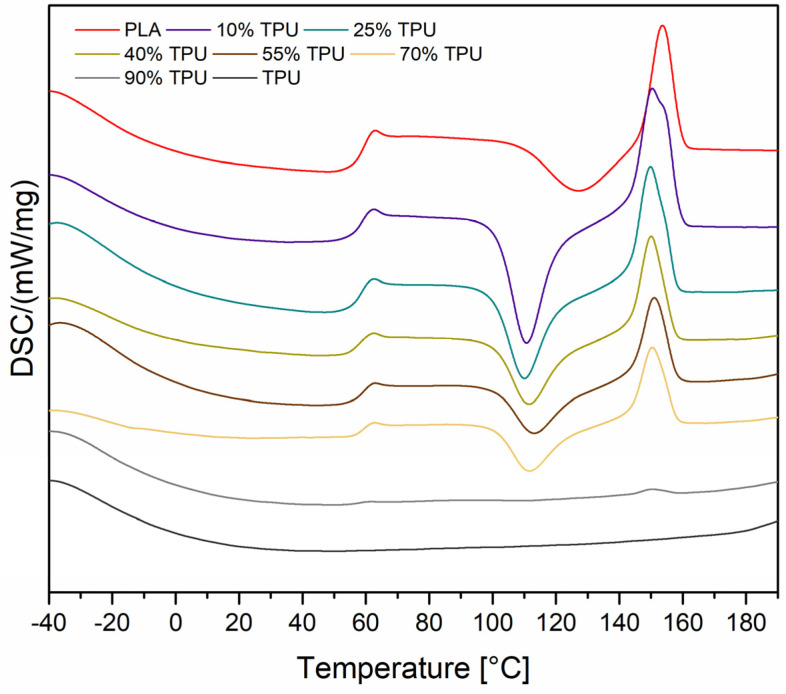
DSC curves recorded for the second heating cycle.

**Table 1 materials-16-06927-t001:** Comparison of the time performance of high-throughput (robot-operated) and traditional three-point bending measurements.

Operator Type	Weekly Working Time ^1,2^ [h]	Single Measuring Time [h]	Number of Samples Tested per Hour [pcs]	Sample Mounting Time [h]
human	30	0.10	250	0.02
robot	138	0.10	1340	0.003

^1^ Maintenance time of the robot = 6 h per week. ^2^ Daily man–machine time/5 days per week.

**Table 2 materials-16-06927-t002:** Injection molding parameters.

**Temperature (°C)**	**Nozzle**	**Zone 3**		**Zone 2**	**Zone 1**		**Feed**
	**210**	**205**		**205**	**200**		**40**
**Mold temperature (°C)**	25						
**Holding pressure**	t (s)		0			11	
	p (bar)		700			1100	
**Clamping force (kN)**	**Holding** **pressure time (s)**		**Cooling time (s)**			**Screw diameter (mm)**	
800	11		60			25	

**Table 3 materials-16-06927-t003:** Results of thermogravimetric analysis (N_2_).

	T_5%_ [°C]	1st Stage		2nd Stage	
		T_1onset_ [°C]	[°C]	T_2onset_ [°C]	T_2max_ [°C]
PLA	309.0	330.5	343.0	-	-
25% TPU	301.5	301.8	333.2	-	353.0
40% TPU	300.8	309.3	326.5	356.8	366.1
70% TPU	296.0	301.7	315.9	343.0	370.3
TPU	293.5	309.9	338.7	382.8	398.4

**Table 4 materials-16-06927-t004:** DSC analysis results.

	T_g_ [°C]		T_cc_ [°C]		T_m_ [°C]	
Cycle	First	Second	First	Second	First	Second
PLA	59.9	61.9	115.9	127.6	154.9	153.8
10% TPU	59.9	61.5	107.6	110.6	155.4	149.9/154.8
25% TPU	60.7	61.6	106.8	109.7	150.9	149.3
40% TPU	60.7	61.6	109.5	111.5	151.3	149.7
55% TPU	60.5	62.0	107.2	112.6	149.8	151.0
70% TPU	59.4	61.4	107.4	111.6	150.3	150.0
90% TPU	-	-	-	-	-	-
TPU	-	-	-	-	-	-

## Data Availability

Not applicable.

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
