# Peer review of "Robotization of Three-Point Bending Mechanical Tests Using PLA/TPU Blends as an Example in the 0–100% Range"

_materials, 2023, doi:10.3390/ma16216927_

Round 1

Reviewer 1 Report

First of all, I would like to thank you very much for choosing our journal for your article. It is a very successful and meticulously prepared article. If you answer the questions I have asked, I would like to read the article again.

- How was it ensured that the mold temperature remained at room temperature? Were there any controls or monitoring systems in place?

- How were the TPU concentrations in the polymer blends verified, especially given the continuous feed system?

- How did the authors ensure uniform heating during the DSC and TG measurements given the specific temperature ranges and heating rates?

- For contact angle measurements, were there any controls in place to ensure that each 5 μl water drop was consistent in size and shape?

- How was the "ready signal" generated? Was there a feedback loop between the testing machine and the Dobot Magician manipulator?

- Can you explain the functionality and design of the "appropriately shaped gripper" that pushed out used samples?

- What were the criteria for determining a successful test, and how were anomalies or outliers dealt with in the dataset?

- How does the setup and methodology in this study compare with other similar studies in the literature? What makes this approach unique or advantageous?

- Could you clarify the mechanisms that lead to the appearance of rounded-edge micro-cracks due to a loss of fluidity between the PLA and TPU micro-areas? Are these related to stress concentrations or material incompatibilities?

- How do the observed microstructural changes in the PLA/TPU blends influence the mechanical properties of the samples?

- How were the macroscopic images in Figure 8 related to the microstructural findings? Were there any discernible patterns or trends?

- Could you provide details on the statistical methods used to determine the trend curve and the coefficient of determination R^2 for the contact angle measurements?

- To select suitable thermoplastic materials these two studies must be in your literature section

Ordu M, Der O. Polymeric Materials Selection for Flexible Pulsating Heat Pipe Manufacturing Using a Comparative Hybrid MCDM Approach. Polymers. 2023; 15(13):2933. https://doi.org/10.3390/polym15132933

Materials selection of thermoplastic matrices for ‘green’ natural fibre composites for automotive anti-roll bar with particular emphasis on the environment

https://doi.org/10.1007/s40684-018-0012-y

- Were there any studies conducted to assess the durability or stability of the hydrophobic properties of the blends over time or under varying environmental conditions?

- Did you observe any significant differences in the DSC results between the first and second measurement cycles, and if so, what do you attribute these changes to?

- How does the reduction in Tcc value with increasing TPU content influence the mechanical and thermal properties of the blends? Are there potential applications where a specific blend ratio might be more advantageous due to these thermal properties?

- In the blends with 90% TPU, why are the Tg, Tcc, and Tm signals barely discernible? Is this attributed to a dominant TPU phase or other factors?

- How do the findings of this study align or diverge from previous literature on PLA/TPU blends, particularly in terms of crystallization behavior?

- Could you further discuss any potential applications or industries that might benefit from these specific blends given their unique thermal and surface properties?

Author Response

Dear reviewer,

Thank you for taking the time to review our paper. We appreciate your valuable feedback and suggestions that will help us improve the quality of our published article. Please find below our responses to the questions raised about the reviewed paper.

  1. How was it ensured that the mold temperature remained at room temperature? Were there any controls or monitoring systems in place?

The mould mounted on the injection moulding machine has a special temperature control system, which is set on the chiller.

  1. How were the TPU concentrations in the polymer blends verified, especially given the continuous feed system?

The material was weighed in appropriate portions (with an assumed concentration) and fed into the injection moulding machine hopper in 1% increments. The stated concentration is a prediction, as we currently do not have such accurate tools to help us measure the concentration with 100% certainty. This is the first phase of testing and we are working to refine it. Ultimately, the concentration will be controlled using spectroscopic methods.

  1. How did the authors ensure uniform heating during the DSC and TG measurements given the specific temperature ranges and heating rates?

The NETZSCH apparatus used for the measurements ensures strict control over the course of the thermal tests. The heating speed used ensures highly accurate measurements.

  1. For contact angle measurements, were there any controls in place to ensure that each 5 μl water drop was consistent in size and shape?

The apparatus used for the measurement provides a fixed droplet size, which is set in the software dedicated to the apparatus. The software provides a preview of the droplet on the surface under test.

  1. How was the "ready signal" generated? Was there a feedback loop between the testing machine and the Dobot Magician manipulator?

There was a feedback loop between the testing machine and the manipulator. Dobot has external pins to send and receive signals. The manipulator picks up a specimen from the automated specimens stack, goes to a safe position near the testing machine, and waits for the signal. The testing machine sends a “ready signal” to the waiting manipulator, so the robot places the specimen in the machine, goes to a safe place, and sends a signal to start the test. This was done in a loop, which the manipulator counted.

  1. Can you explain the functionality and design of the "appropriately shaped gripper" that pushed out used samples?

The gripper in this application works vertically (takes specimens from the side) and has a lower finger longer than the upper so that it can push the tested specimen behind the machine after the test. Specimens were numbered so they wouldn’t get mixed. Also, the gripper had a step on which the specimen rested during collection, so it was always in the same repetitive position.

  1. What were the criteria for determining a successful test, and how were anomalies or outliers dealt with in the dataset?

The testing machine works until specimens are in the automated specimens stack. Each specimen batch counted 100 pieces. After each test, the machine saves data to a file. Figures with force-displacement path were checked and analyzed after the feeder had been empty. If there were doubts, the figures/data were compared with the specimen and checked if it was correctly tested. If the test results showed a deviation from the average level of 15%, the data were rejected.

  1. How does the setup and methodology in this study compare with other similar studies in the literature? What makes this approach unique or advantageous?

No similar literature data was found for the continuous characterisation of polymer blends and the use of a similar test bench as ours to carry out 3-point bending in automatic mode. High-throughput experiments can explore materials parameters across endless composition combinations, supporting mathematical calculations or preparing data libraries for AI tools. Despite potential benefits, this experimentation and research planning approach is met with academic resistance due to the perception that it's inadequate for fundamental research, involves significant financial expenditure, and is unsuitable for complex materials with physicochemical properties. Transitioning from a traditional one-at-a-time experimentation model to the use of advanced tools requires systematic familiarization with the new experiential approach and implementation of simple high-throughput research tools, especially in academic institutions. This allows for the training of qualified personnel who can efficiently operate complex research systems, requiring knowledge and skills from multiple disciplines. The use of HTE and HTT is an essential step towards creating an environment for the development of self-learning artificial intelligence (AI) networks and inter-intelligent, self-managing machine-to-machine (M2M) systems. This is particularly relevant in fields like drug design, biomaterials, biology, and biotechnology.

  1. Could you clarify the mechanisms that lead to the appearance of rounded-edge micro-cracks due to a loss of fluidity between the PLA and TPU micro-areas? Are these related to stress concentrations or material incompatibilities?

The observed microcracks and microstructural changes are related to the stress concentration resulting from the direction of force application during the test. As the TPU content of the blends increases, the value of the modulus of elasticity changes, so the microstructural damage caused by the external force is significantly less.

  1. How do the observed microstructural changes in the PLA/TPU blends influence the mechanical properties of the samples?

The observed changes in microstructure are the result of changes in the mechanical properties of the materials tested, in particular, a decrease in the degree of micro-damage is caused by a decrease in the modulus of stiffness.

  1. How were the macroscopic images in Figure 8 related to the microstructural findings? Were there any discernible patterns or trends?

These are macroscopic images of chosen samples are presented. The first one is neat PLA and the last one is neat TPU.

The test material's characteristics change from the typically brittle failure due to the effects of bending stresses observed in PLA (complete failure of the specimen) to an elastic response due to stress relaxation after force subtraction and return to the original form for test objects made of TPU (lines 306-309 in the manuscript, highlighted in gray).

  1. Could you provide details on the statistical methods used to determine the trend curve and the coefficient of determination R^2 for the contact angle measurements?

The fitting of the trend curve was carried out using the analytical tools available in OriginPro 2016 software. The data obtained from the measurements were entered into the software, and a linear regression model was selected. The software automatically fitted the regression model to the data entered, finding the values of the slope coefficient of the curve (slope) and the Y-axis intersection point (intercept) that best matched the data entered. After fitting the model, the program calculated the values predicted by the model for all points, the differences between the actual data values, and the values predicted by the model. Subsequently, the sum of the squares of these differences was calculated, which is called the Sum of Squares of Errors (SSE) and the Sum of Total Squares (SST), which is the sum of the squares of the differences between the actual data values and their mean. From the SSE and SST values, Origin Pro calculated the coefficient of determination R^2 using the formula: R^2 = 1 - (SSE / SST).

The statistical methods for both WCA and three-point bending tests have been added to the manuscript in the materials and methods section (highlighted in yellow, lines 205-218).

  1. To select suitable thermoplastic materials these two studies must be in your literature section

Ordu M, Der O. Polymeric Materials Selection for Flexible Pulsating Heat Pipe Manufacturing Using a Comparative Hybrid MCDM Approach. Polymers. 2023; 15(13):2933. https://doi.org/10.3390/polym15132933

Materials selection of thermoplastic matrices for ‘green’ natural fibre composites for automotive anti-roll bar with particular emphasis on the environment https://doi.org/10.1007/s40684-018-0012-y

Thank you for your kind advice. These two studies have been added to the literature section of the manuscript and highlighted in yellow (lines 85-98, bibliography positions 17 and 18).

  1. Were there any studies conducted to assess the durability or stability of the hydrophobic properties of the blends over time or under varying environmental conditions?

At this stage of the study, no tests were carried out to determine the stability of the hydrophobic-hydrophilic properties over time, nor were environmental tests conducted.

  1. Did you observe any significant differences in the DSC results between the first and second measurement cycles, and if so, what do you attribute these changes to?

The differences between the signals obtained in the first and second heating cycles are described in the text of the manuscript (lines 356-364, highlighted in gray). The changes in characteristic temperatures are summarised in Table 3.

„The DSC measurements conducted during the second cycle indicate that the presence of dispersed TPU phases had an impact on the PLA crystallization process. The Tcc value decreased from almost 128°C (neat PLA) to around 110°C (PLA/TPU blend). The thermograms illustrate that the cold crystallization peak is more pronounced for the blends, especially for those containing 10% and 25% TPU. According to the literature, this effect can be attributed to TPU acts as a crystallization nucleation agent by providing nucleation spots. In blends that contain 90% TPU, the Tg, Tcc, and Tm signals are barely discernible”.

  1. How does the reduction in Tcc value with increasing TPU content influence the mechanical and thermal properties of the blends? Are there potential applications where a specific blend ratio might be more advantageous due to these thermal properties?

A decrease in Tcc may have the effect of increasing the degree of ordering in the blende, which in turn may result in an increase in the degree of crystallinity, which translates, for example, into a decrease in the degree of transparency of the material, which may therefore act as a barrier to radiation, and such materials could therefore be used in the manufacture of packaging.

  1. In the blends with 90% TPU, why are the Tg, Tcc, and Tm signals barely discernible? Is this attributed to a dominant TPU phase or other factors?

This attributed to a dominant TPU phase.

  1. How do the findings of this study align or diverge from previous literature on PLA/TPU blends, particularly in terms of crystallization behavior?

The reference to the literature data is described in the text of the manuscript (lines 340-343, highlighted in gray). According to the literature, this effect can be attributed to TPU acts as a crystallization nucleation agent by providing nucleation spots. The cited paper is reproduced below:

Mi, H.Y.; Salick, M.R.; Jing, X.; Jacques, B.R.; Crone, W. C.; Peng, X.F.; Turng, L.S. Characterization of thermoplastic polyurethane/polylactic acid (TPU/PLA) tissue engineering scaffolds fabricated by microcellular injection molding. Materials Science and Engineering: C 2013, 33(8), 4767–4776. doi:10.1016/j.msec.2013.07.037

  1. Could you further discuss any potential applications or industries that might benefit from these specific blends given their unique thermal and surface properties?

It can be probably used in many potential applications for packing industries, 3D-printed objects, cup mats, toys, and so on.

Reviewer 2 Report

General comment:

The article discusses the automation of the mechanical test using PLA/TPU blends as an example. the paper is well structured. it can be further improved by addressing the following concerns.

Specific comments:

1. Typo in abstract, should be "throughput" in line 16.

2. i suppose this is a fully automated process. however, from figure 1, it is not clear to me how the samples created from the injection moulding will be feed to the conveyor system. 

3. i thought the main focus and the novelty of this study is on the robotization of the flexural test, based on my understanding from the title. however, the section 3.3 and 3.4 are not performed using the robot. 

4. it would be interesting to know if in-situ video capturing is also performed to correlate the crack formation to the trend in the response curve.

5. suggest discussing whether the other tests such as DSC, contact angle analysis, and microstructure evolution requires the use of high-throughput method?

6. i would expect more investigation into the statistical significance and the confidence level of the results between the traditional ones and the automated ones, keeping time as the constant. i.e. for the same amount of time, the how many tests can human or the robots perform, and how does it affect the statistical significance.

7. figure 8, label the respective specimens in terms of the compositions.

8. figure 6, the font size of the axis title can be larger for better readability. the choice of the color may not be the best, consider using other method such as line patterns or gray scale.

9. figure 3, the pictures in the figure look messy, probably due to the poor contrast. suggest improving the picture quality such that it is easier to tell what you are trying to show in each picture.

10. characterization of mechanical properties is important in material development, including 3d printing processes which often entails anisotropic materials properties, making it more sense to use the automation of the mechanical tests. suggest citing:

a. Goh, G. D., Toh, W., Yap, Y. L., Ng, T. Y., & Yeong, W. Y. (2021). Additively manufactured continuous carbon fiber-reinforced thermoplastic for topology optimized unmanned aerial vehicle structures. Composites Part B: Engineering216, 108840.

b. Chen, J., Liu, X., Tian, Y., Zhu, W., Yan, C., Shi, Y., ... & Zhou, K. (2022). 3D‐Printed anisotropic polymer materials for functional applications. Advanced Materials34(5), 2102877.

c. Ma, G., Li, Z., Wang, L., Wang, F., & Sanjayan, J. (2019). Mechanical anisotropy of aligned fiber reinforced composite for extrusion-based 3D printing. Construction and Building Materials202, 770-783.

1. Typo in abstract, should be "throughput" in line 16.

Author Response

Dear reviewer,

Thank you for taking the time to review our paper. We appreciate your valuable feedback and suggestions that will help us improve the quality of our published article. Please find below our responses to the questions raised about the reviewed paper.

  1. Typo in abstract, should be "throughput" in line 16.

It has been corrected (highlighted in yellow).

  1. I suppose this is a fully automated process. however, from figure 1, it is not clear to me how the samples created from the injection moulding will be feed to the conveyor system. 

The injection-moulded samples will, after being pushed out of the mould, fall by gravity onto a conveyor with which they will be transferred to the testing machine.

  1. I thought the main focus and the novelty of this study is on the robotization of the flexural test, based on my understanding from the title. however, the section 3.3 and 3.4 are not performed using the robot. 

The main key of the paper is the application of the designed test bench to the characterisation of polymer blends, but in addition to this, the paper presents the characterisation of the developed materials, which is covered in the sections listed. In the future, such measurements can also be automated and robotised.

  1. It would be interesting to know if in-situ video capturing is also performed to correlate the crack formation to the trend in the response curve.

No video capturing were performed in these studies.  This type of activity is planned for future studies using a high-speed imaging camera

  1. Suggest discussing whether the other tests such as DSC, contact angle analysis, and microstructure evolution requires the use of high-throughput method?

The indicated test methods can be carried out using high-throughput methods, but the construction of suitable test benches requires significantly more money and time to prepare. In the future, however, it may be possible to make such adaptations, for existing equipment. For DSC or TGA measurement instruments, tools are available on the market to carry out measurements in these modes.

  1. I would expect more investigation into the statistical significance and the confidence level of the results between the traditional ones and the automated ones, keeping time as the constant. i.e. for the same amount of time, the how many tests can human or the robots perform, and how does it affect the statistical significance.

A broader discussion and comparison is set out in the table below. The table is included in the manuscript.

  1. figure 8, label the respective specimens in terms of the compositions.

Figure 8 and its description (highlighted in yellow) have been corrected.

  1. figure 6, the font size of the axis title can be larger for better readability. the choice of the color may not be the best, consider using other method such as line patterns or gray scale.

Thank you for your kind note about the graphs. Below are various proposals for modified diagrams. Which one should we include in the text of the manuscript? Maybe the solution will be to add bigger diagrams in the firstly used pallet of colours?

  1. figure 3, the pictures in the figure look messy, probably due to the poor contrast. suggest improving the picture quality such that it is easier to tell what you are trying to show in each picture.

Thank you for your kind advice. In the drawing, we wanted to represent the overall work of the stand. When zoomed in, the pictures are clearly visible. Due to the purpose of the introduced graphic, we will not be subjecting it to further modification.

  1. characterization of mechanical properties is important in material development, including 3d printing processes which often entails anisotropic materials properties, making it more sense to use the automation of the mechanical tests. suggest citing:

    a. Goh, G. D., Toh, W., Yap, Y. L., Ng, T. Y., & Yeong, W. Y. (2021). Additively manufactured continuous carbon fiber-reinforced thermoplastic for topology optimized unmanned aerial vehicle structures. Composites Part B: Engineering216, 108840.

    b. Chen, J., Liu, X., Tian, Y., Zhu, W., Yan, C., Shi, Y., ... & Zhou, K. (2022). 3D‐Printed anisotropic polymer materials for functional applications. Advanced Materials34(5), 2102877.
  2. Ma, G., Li, Z., Wang, L., Wang, F., & Sanjayan, J. (2019). Mechanical anisotropy of aligned fiber reinforced composite for extrusion-based 3D printing. Construction and Building Materials202, 770-783.

Thank you for your kind advice. Your propositions have been added to the manuscript text and highlighted in yellow (lines 56 – 58, bibliography position 14-16).

Round 2

Reviewer 2 Report

the replies are satisfactory. the quality of the manuscript has improved in the revised version.